# Outcome Evaluation of Distal Femoral Fractures Following Surgical Management: A Retrospective Cohort Study

**DOI:** 10.3390/jpm13020350

**Published:** 2023-02-17

**Authors:** Mirjam V. Neumann-Langen, Verena Sontheimer, Gudrun H. Borchert, Kaywan Izadpanah, Hagen Schmal, Eva J. Kubosch

**Affiliations:** 1Klinikum Konstanz, Department of Orthopaedic and Trauma Surgery, Mainaustrasse 35, 78464 Konstanz, Germany; 2Department of Orthopedics and Trauma Surgery, Medical Center—Albert-Ludwigs-University of Freiburg, Hugstetter Strasse 55, 79106 Freiburg, Germany; 3Dr. Borchert Medical Information Management, Egelsbacher Strasse 39e, 63225 Langen, Germany; 4Department of Orthopedic Surgery, University Hospital Odense, Sdr. Boulevard 29, 5000 Odense, Denmark

**Keywords:** distal femur, femoral fracture, locking plates, screw fixation, complications, longitudinal axis deviation

## Abstract

Background: Distal femur fractures are challenging in surgical management as the outcome is crucial for restoring the biomechanical stability and longitudinal axis of the leg and function of the knee joint. Methods: A retrospective review of all distal femoral fractures treated in a level I trauma center over a decade was performed. The radiographs were reviewed for fracture entity, osseous healing, implant failure, mechanical axis, and degenerative joint changes. Clinical outcome was reviewed regarding postoperative complications and postoperative range of motion of the knee joint. Results: 130 patients who were managed with screw fixation (*n* = 35), plating systems (*n* = 92) or intramedullary nailing systems (*n* = 3) remained for evaluation. Mean follow up was 26 months. Clinical outcome was significantly better for flexion degrees following screw fixation (*p* = 0.009). Delayed fracture union (*p* = 0.002) or non-union (*p* = 0.006) rates were significantly higher in plate osteosynthesis. Mild pathologic deformity for varus and valgus collapse was found following plate osteosynthesis. Conclusions: Screw fixation shows fewer postoperative complications than plate fixation and is favored for extra and partial intraarticular distal femur fractures. Plating constructs remain the superior fixation method in complex distal femur fractures but are associated with higher rates of non-union and leg axis deviation.

## 1. Introduction

Fractures of the femur represent about 3–6% of all musculoskeletal injuries, with the distal femur being involved in about 1% of all cases [1]. Injuries to the femur follow a bimodal distribution, resulting from high-energy trauma, such as traffic accidents, in young patients and low-energy trauma, such as tripping, in older patients [2]. The incidence of distal femur fractures is highest in females >75 years old and in adolescent males 15–24 years old [2].

Since the distal femur plays a crucial part in the biomechanical functionality of the knee joint, in addition to in the longitudinal axis stability of the leg, the surgical treatment of these fractures is vital for the mobilization and resilience of the patient. Common deformities in distal femur fractures include shortening, flexion and external rotation of the proximal fragments, and the extension of the distal fragments. These issues are the result of powerful muscles, such as the gastrocnemius and the adductor muscle, which insert on and exert unilateral forces on the distal femur [3].

With the expected dynamic demographic change and the more active lifestyle of older people, it is likely that injuries to the distal femur will increase as well. Non-operative treatment achieved acceptable results back in the early 1960s [4]. During the 1990s to the early millennium, a great deal of attention was paid to implant development and the comparison of different implant types for surgical management of distal femur fractures. Thus, numerous publications appeared that compared different plating systems for simple transverse or complex intraarticular distal femoral fractures [4,5,6,7,8,9]. The various plating systems, such as blade plates, dynamic compression plates, or locking compression plates, are all suitable for extraarticular, sagittal unicondylar, or supra- and intercondylar distal femur fracture types. Several biomechanical investigations showed the superiority of locking compression plates compared to classic internal fixation (DCP plate, retrograde nailing, blade plate) [10].

Some fracture types are suitable for surgical treatment with an intramedullary force carrier. The improvement of the nail design with a retrograde knee insertion point has given this procedure a relevant status. Biomechanically, the intramedullary nail shows greater axial stability and fewer micromovements compared to dynamic condyle screws and locking condyle plates [11]. However, intramedullary nailing systems are difficult in comminuted metaphyseal fractures with coronal plane involvement. Since clinical and functional results remain conflicting, the improvement of the characterization of the prognosis of these surgical treatments is critical.

The aim of the retrospective cohort study is the critical analysis of clinical and radiographic outcome of different surgical methods in the management of distal femoral fractures. The comparison of different osteosynthesis procedures based on the incidence of postoperative complications allows a prediction of better treatment methods.

To our knowledge, isolated small or large fragment screw fixation, except in the management of Hoffa fractures, has not yet been further evaluated. We also aim to clarify which surgical approach has the fewest complications for the surrounding tissue and the patient and the potential postoperative deviation of the longitudinal femur axis. In addition, it should also be investigated whether intraarticular fracture involvement has a relevant influence on the development of posttraumatic degenerative changes to the knee joint.

## 2. Materials and Methods

The clinical database of a level I trauma center was searched for surgically managed distal femoral fractures using the postoperative diagnosis variable codes S72.40–S72.44 (International Classification of Diseases (ICD, 10th Revision)). The data acquisition period was over 10 years between 1 January 2010 and 31 December 2020. Inclusion criteria for this retrospective cohort study were complete clinical and radiographic patient charts, surgically managed fractures using plating system, screws, or intramedullary nail, patient’s age at time of surgery > 18, and postoperative follow up at the outpatient clinic of a minimum of 3 months. Exclusion criteria were juvenile fractures and periprosthetic and pathological fractures.

Each patient record was reviewed for patient factors (gender and age) and comorbidities (e.g., smoking history, history of diabetes, obesity defined as BMI > 35), injury-related factors (soft tissue damage according to the Gustilo–Anderson classification [12]), the fracture entity using the AO-ASIF classification [13], and development of infection. The surgical method of fracture management was either using variable angle locking or non-locking plates, cannulated small or large fragment screws, cancellous or cortical screws, or intramedullary nailing. The surgical approach was reviewed and documented to allow analysis of correlation between surgical approach and postoperative complications. Postoperative complications were considered as early if they occurred within 14 days after surgery, such as postoperative hemorrhage, wound infection, or thromboembolism. Late postoperative complications were recorded with secondary fracture dislocation, implant failure, delayed fracture union or fracture non-union, and leg axis deviation. Radiographic union was defined as the presence of a minimum of three out of four bridging cortices on plain anteroposterior and lateral radiographs [14]. Determination of the anatomical and mechanical axes and joint angles in the frontal plane of the full leg radiograph was analyzed according to the Paley measurement [15]. During the postoperative course of 6 weeks, 3 months, 12 months, and latest follow up in the outpatient clinic, the postoperative range of motion (ROM), as well as postoperative degenerative joint changes according to the Kellgren–Lawrence classification [16] on plain radiographs in comparison to the pre-surgery status, was documented using the neutral zero method. The presence of radiographic signs of osteoarthritis was looked for 6 months postoperatively at the earliest. The study cohort was divided into three groups according to the technique of surgical management: screw fixation (group 1), plate osteosynthesis (group 2), or intramedullary nail (group 3).

Statistical analysis was performed with OriginPro, version 2023 (OriginLab Corporation, Northampton, MA, USA). Due to non-normal distribution of the values, a nonparametric test was used. For continuous values, Kruskal–Wallis ANOVA was used with Dunn’s test for comparison of more than 2 groups. For ordinal data, the chi-square test was used. A *p*-value < 0.05 was considered to indicate statistical significance. Due to the exploratory character of the study, no correction was made for multiple testing; calculated *p*-values are purely descriptive.

The study protocol was approved by the local Human Research Ethics Committee of the Albert Ludwig University of Freiburg, ethical approval code: 22-1287-S1-retro.

## 3. Results

Within the data acquisition period of 10 years (2010–2020), 520 patients were identified who sustained distal femoral fractures. Considering the specific inclusion criteria, 130 patients remained for the retrospective data analysis (Figure 1).

There were 71 male patients and 59 female patients; mean age at the time of surgery was 52.3 ± 20.5. Mean follow up was 25.8 months (3–170 months) in this study cohort. Relevant comorbidities were found in 92/130 included patients. There were 29 patients suffering from a bone metabolism disorder such as osteoporosis or osteopenia, 17 had a cardiovascular history, 14 had obesity defined by a BMI > 35, 11 had a diabetic history, and 6 were smokers. However, a statistical significance between the primary union group and the non-union group for those with a bone metabolism disorder was recorded (Table 1).

The included patients were allocated to groups with respect to the surgical management: screw fixation using either small or large fragment screws (group 1, 35 patients), variable angle plate or locking vs. non-locking plate osteosynthesis (group 2, 92 patients), or intramedullary nailing system (group 3, 3 patients) (Figure 2).

Hence, statistical analysis was carried out between groups 1 and 2. Three patients in group 1 sustained bilateral distal femoral fractures; 13 bilateral distal femoral fracture injuries were found in group 2. A tabular overview of demographic data, fracture entities, and postoperative complications is given in Table 2.

In group 2, the LISS plate (less invasive stabilization system, DePuy Synthes, Oberdorf, Switzerland) was used in 78 patients; 10 patients were managed using a VA-LCP; a T-shaped LCP was used in two patients; one patient received an NCB (non-contact bridging plate, Zimmer Biomet, Zug, Switzerland); and one patient received a TomoFix plate (DePuy Synthes, Oberdorf, Switzerland). Combined fracture management using a plate system and additional fragment screw fixation was performed in 73 out of 92 patients (80%).

Simple extraarticular or extraarticular multifragmentary distal femoral fractures AO 33A1-3, as well as intraarticular distal femoral fractures AO 33C1-3 (*p* < 0.0001), were significantly more often managed with plating systems. Partial articular distal femoral fractures, as well as unilateral condyle fractures, were predominantly fixed with screws. Further sub-analysis of the correlation between the fracture entity and fracture union was performed for group 2 as the included number of patients allowed statistical analysis. Wound healing disorder during the early postoperative course was significantly more often seen following AO type 33C2 and C3 (*p* = 0.032). Osseous non-union > 9 months post-surgery was higher in AO fracture type 33C1-3 (*p* = 0.049) (Table 3).

Soft tissue damage, including open fractures of grades II and III according to the Gustilo Anderson classification [12], was more frequently seen in group 2 without any significance between groups. Two-step surgeries with initial closed reduction and external fixation were seen significantly more often in group 2 (*p* = 0.022). Early postoperative complications such as wound healing problems, wound infection, or thromboembolism did not show any significance between groups (Table 1). Of the reviewed patients, 9% (3/32 patients with available data) in group 1 suffered early wound healing disorders <14 days after surgery and 18% (14/78 patients with available data) in group 2. The infection rate over 6 weeks post-surgery was 4% (1/24 patients with available data) in group 1 and 22% (11/49 patients with available data) in group 2. Late complications such as delayed fracture healing 3–9 months postoperatively were found in one patient in group 1 and in 29 patients in group 2 (*p* = 0.002). Fracture non-union >9 months postoperatively was found significantly more often following plate osteosynthesis (*p* = 0.006), 25 patients in group 2 versus one patient in group 1. The osseous non-union rate during the early postoperative course of 6 weeks was significantly higher in group 2 (*p* = 0.023); further follow up showed no significant differences in the bony healing phase between the examined groups, with completed osseous healing at latest follow-up >2 years (Figure 3).

A detailed review of osseous union versus non-union in group 2 showed that, first, the patient’s age had a significant influence on the osseous healing (*p* = 0.011), with patients aged 31–50 years tending to experience osseous non-union; second, the soft tissue damage was significant for the bony healing (*p* = 0.027); and, third, bacterial infection to the soft tissue occurred during the postoperative course (*p* = 0.001). Seven of twenty-five patients were affected with infectious osseous non-union. In all cases, soft tissue damage of Gustillo–Anderson classification grade 2 or higher was documented. This was managed with complete removal of the plating system and hybrid external fixation system and revision surgery after secured restoration of the infection. Two of these patients received an additional fibula graft. Osseous healing was achieved within 16 months (12–30 months) post-revision-surgery. An atrophic osseous non-union was found in 15 patients; these patients were managed with revision plate surgery and additive cancellous bone. Osseous healing was completed after a mean of 12 months (8–16 months). Three other patients affected with atrophic non-union were managed with exogen therapy.

The surgical approach was chosen depending on the choice of implant. When using plating systems, the anterolateral approach was the most common (71/92 patients). Parapatellar and medial, but also percutaneous accesses, were chosen most frequently for screw osteosynthesis. The surgical approach did not show any impact on the complication rate (*p* > 0.999 between groups and *p* = 0.595 for union vs. non-union rate in plate group).

Postoperative degenerative joint changes according to the Kellgren–Lawrence classification [16] showed an increase in degenerative posttraumatic joint changes in conventional radiographic examination of at least one Kellgren-Lawrence classification grade at 6 months post-surgery vs. pre-surgery and >1 year post-surgery vs. pre-surgery in both groups without relevant statistical significance. Postoperative range of motion was significantly better in the flexion ranges in group 1 (*p* = 0.009), with flexion rates of 122.50 ± 14.64 degrees vs. 104.74 ± 26.12 in group 2 (Figure 4).

There was no knee flexion deficit for the range of motion <90° in either group, but there was a loss of extension >10° in 6 of 57 patients with the available data in group 2 at the latest follow up. Implant malalignment, defined as loss of the bony contact surface to the plate or incorrect positioning of the screws, was documented significantly more often in group 2 (*p* = 0.002).

A total of 35 patients were reviewed for postoperative mechanical leg axis development (group 1, *n* = 10; group 2, *n* = 25). The analysis of the mechanical leg axis showed no statistically significant differences between groups, but mild or pathologic deformities were found more often in group 2 compared to in group 1 (Table 4).

A mild pathologic deformity with varus collapse was found in 7 out of 25 (28%) reviewed patients with available data in the plating group, as well as a mild valgus collapse in 15/25 (60%) patients in the same group without any significance between groups. Four out of sixty-five patients in group 2 had a pathologic leg length discrepancy of >1.5 cm at the latest follow up.

## 4. Discussion

The aim of the presented retrospective cohort study was to determine the clinical and radiographic long-term outcome of different methods of surgical management of distal femur fractures. Simple extraarticular or partial intraarticular distal femoral fractures were predominantly fixed with screws with evidently fewer postoperative complications. The clinical outcome was significantly better for the postoperative range of flexion degrees after screw fixation. Osseous union was completed earlier after screw osteosynthesis. Delayed osseous union and non-union rates were affected by the patient’s age, the sustained soft tissue damage, and infection during the postoperative course, as well as comminuted fracture types. Mild pathological changes affecting the longitudinal axis of the leg were found in 28% with varus collapse and 60% with valgus collapse after surgical fracture treatment with plate systems. The chosen surgical approach had no impact on the postoperative course regarding infection or implant misalignment. Posttraumatic degenerative changes with radiographic changes showed a progression in each reviewed group without statistical significance.

Two-step surgery with initial placement of an external fixator was significantly more often performed prior to definite plate fixation of distal femur fractures. These, in turn, were used more frequently in complex intraarticular AO 33C1-3 distal femoral fractures. Further sub-analysis of the fracture entity showed that comminuted fracture types AO 33C2 and C3 significantly more often had wound healing problems during the early postoperative course.

The indication for intramedullary nail or plate fixation in fracture management of distal femur fractures is dependent on many factors: the degree of comminution, coronal plane involvement, bone quality, and distal extent of the fracture. As predominantly intraarticular distal femur fractures were reviewed in the presented study, the indication for fracture management using an intramedullary nailing system was very limited. The fixation of distal femur fractures using screws only was most frequently discussed following unicondylar fracture types. In our cohort, only 10 patients had a defined Hoffa fracture type entity AO 33B3.2.

Only a few studies were available for direct comparison with the presented findings.

A retrospective cohort study with 116 patients with a mean follow up of 11 years showed that functional outcomes, as well as postoperative complication rates, were significantly superior following condylar screw fixation compared to following plate fixation [17]. In a smaller retrospective cohort analysis with 57 patients included and a three-year follow up, significantly earlier callus formation was found following interfragmentary screw fixation and plate osteosynthesis, as well as significantly earlier average time to full weight bearing after interfragmentary screw fixation [18]. Isolated screw fixation is often reserved for the surgical management of unicondylar distal femur fractures. A mechanical investigation of femoral synthetic composite bones showed that 6.5 mm cancellous screws provide the most rigid fixation [19].

A biomechanical analysis of distal femur fracture fixation in a human cadaveric study showed that axial stiffness and cyclic loading were significant higher following locked compression plate fixation, but no differences were found in torsional stiffness between locked compression plate fixation and dynamic condylar screw fixation [20]. Another biomechanical, in vitro study reached conclusions that were controversial compared to the above-mentioned study and stated that locking plate fixation in distal femur fractures resulted in a stronger construct than dynamic condylar screw fixation in both cyclic loading and ultimate strength of a simulated A3 distal femur fracture [6]. A clinical prospective study reviewed the outcome of 62 patients managed with the less invasive stabilization system (LISS) and stated good functional results with mean flexion degrees of 112–114° and only 2 out of 50 reviewed patients had osseous non-union [21]. In comparison, we had an osseous non-union rate due to wound infection following traumatic soft tissue damage in 28% of patients and atrophic non-union in 60% in the early postoperative follow up. All affected patients were managed with revision surgery. At the final follow up, the remaining non-union rate was 7% in our study, which is in accordance with the comparative data. A retrospective cohort study reviewing 111 patients stated that submuscular plate insertion reduces the non-union rate [22]. Working length of the plate had no impact on outcome parameters.

Dual plating of distal femur fractures has gained more attention for preventing varus collapse and implant failure in comminuted metaphyseal and articular fractures. In a systematic review, satisfactory results were found for comminuted metaphyseal and articular fractures following dual plating, but no differences were found between single lateral plating and dual plating with regards to non-union rate, blood loss, functional outcomes, and complications, although dual plating led to faster fracture healing [23]. A large cohort series of 335 cases reviewed risk factors for failure of locked plate fixation in distal femur fractures and identified the following risk factors for reoperation to promote union and complications: open fracture, diabetes, smoking, increased body mass index, and shorter plate length [24]. Another retrospective review of 283 distal femoral fractures managed with lateral locking plating defined obesity, open fracture, occurrence of infection, and the use of stainless steel as prognostic risk factors of non-union in distal femoral fractures [25]. In our cohort study, pre-existing risk factors and diseases were reviewed, but, except for pre-existing bone metabolism disorders, no other comorbidities were detected that had a relevant influence on the bone healing process. A recent systematic literature review concluded that dual plating constructs are mechanically stronger than other constructs and should be considered for patients with distal femur fractures that have risk factors for instability, varus collapse, or non-union [26].

Two meta-analyses have been published recently that specifically reviewed the occurrence of improper healing and reoperations after different techniques for the surgical stabilization of distal femoral fractures [27,28]. Both papers stated that 5% of all distal femur fractures fail to heal properly, regardless of whether they are treated with a plate or an intramedullary nail. In the presented study, the non-union rate was 7% at the latest follow up following plate osteosynthesis.

More pragmatic surgical treatment approaches have been discussed in the recent past. The management of open type C3 distal femur fractures with primary definitive fixation with a condylar locking plate and an antibiotic-impregnated collagen sheet showed better results in terms of functional and radiological outcomes [29]. Another treatment option that should also be considered, particularly in elderly patients over 85, is the implantation of a megaprothesis. Published data show that distal femur replacement trends towards lower revision and reoperation rates with similar outcomes when compared to other osteosynthetic procedures but has increased estimated blood loss and an extended length of stay in the hospital [30]. A recently published case series of 11 consecutive patients managed with a megaprothesis following distal femur fractures showed a good functional and radiological outcome after a mean follow up of 23 months [31].

The presented results are in accordance with other published data. Screw fixation had superior knee function results at the latest follow up; an extension deficit of >10° was found in 10% (6/57 patients with available data) following plate osteosynthesis. Neither the surgical approach in general nor minimal invasive surgical fixation methods had an impact on postoperative complications. Posttraumatic degenerative joint changes of the affected knee were observed without any significance in occurrence or progression between the reviewed groups. Osseous union rates showed a significant superiority for screw fixation. This information is with respect to less complex fracture entities and less soft tissue damage.

To our knowledge, a specific review of potential changes of the longitudinal femur axis has not yet been presented. Those patients who achieved a long leg radiographic examination at the latest follow up showed a mild pathologic varus collapse of 91–96° in 28% and valgus collapse of 73–78° in 60% of cases following plate osteosynthesis but without statistical significance.

There were some limitations to this study that deserve consideration. The retrospective character of the study limited the precision of the data and means that all surgical treatments and chosen implants were at the discretion of the operating surgeon. The nature of a single-center study limited the achievable data as well. Respecting the eligibility of inclusion criteria led to a high number of excluded patients. As distal shaft fractures were excluded from the presented cohort study, the number of patients who were managed with an intramedullary nailing system was low and could not be considered for statistical analysis. The group size of patients managed with plating systems was almost three times larger than the comparison group of screw fixation. Hence, the statistical outcome was critical to discuss and allowed clinical relevance rather than statistical relevance. Additionally, although we found multiple independent variables that predicted healing complications, such as soft tissue damage or patient’s age, multiple independent variables that predict healing complications were not analyzed (e.g., use of corticosteroids).

## 5. Conclusions

The results of the retrospective cohort study show that fewer complex distal femoral fractures are suitable for screw fixation with superior functional results and a better radiographic long-term outcome. For comminuted distal intraarticular femur fractures, plating systems remain the preferred treatment method, although predictable and non-predictable outcome factors are known. Comminuted distal femur fractures, patient’s age, soft tissue damage, and early wound healing problems are risk factors for osseous non-union. Valgus collapse is a relevant risk factor following plating constructs. Current, available data are consistent in the outcome of plating systems but inconsistent in comparison of contemporary treatment options. Further randomized, controlled trials including patient-related functional outcomes are missing to define evidence-based results.

## Figures and Tables

**Figure 1 jpm-13-00350-f001:**
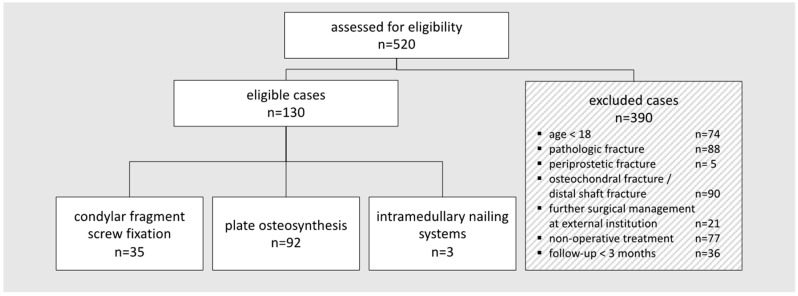
Flow chart calculating the eligibility of included patients.

**Figure 2 jpm-13-00350-f002:**
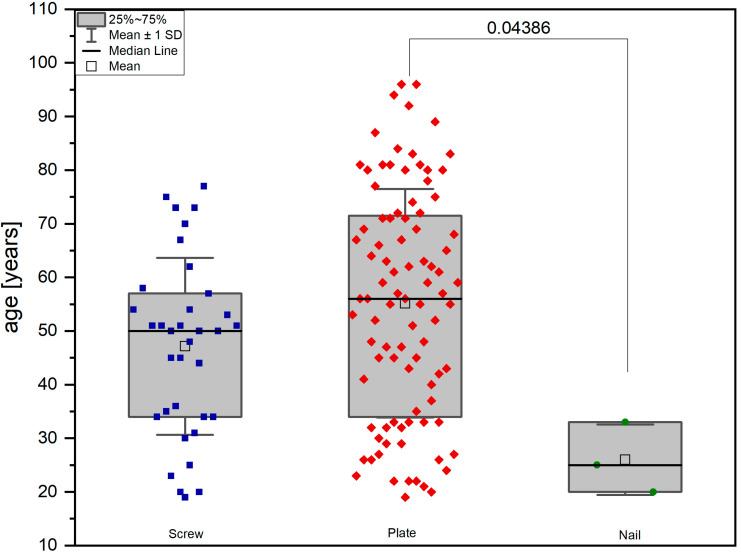
Distribution of age between allocated groups with different fracture management.

**Figure 3 jpm-13-00350-f003:**
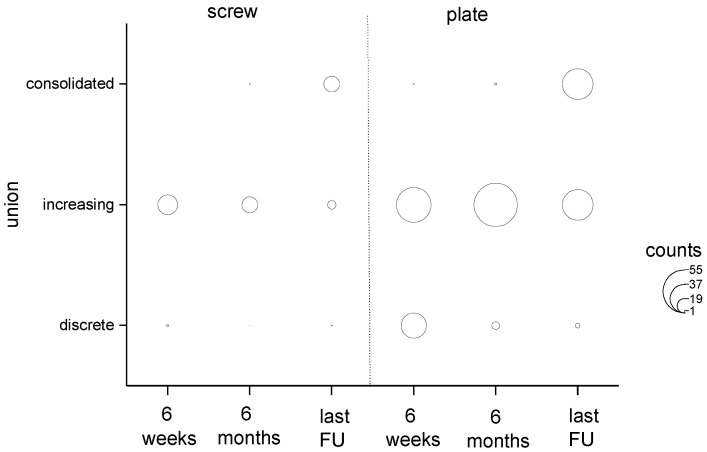
Osseous union time between groups 1 and 2.

**Figure 4 jpm-13-00350-f004:**
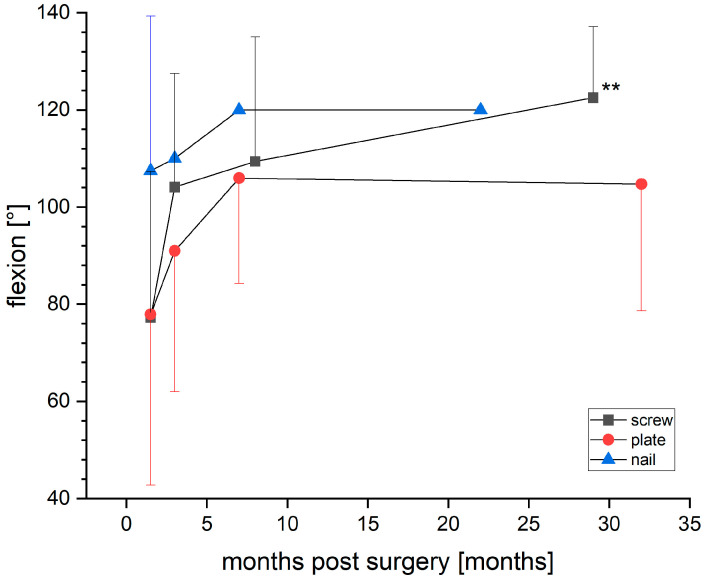
Flexion parameters and outcomes in comparison between groups. ** *p* = 0.009 vs. plate group.

**Table 1 jpm-13-00350-t001:** Correlation between union and comorbidities and risk factors.

Comorbidities and Risk Factors	*n* (%)		*p*-Value
	Primary Union	Non-Union	
Diabetes	5/60 (8)	2/25 (7)	>0.999
Smoker	3/62 (5)	1/26 (6)	>0.999
Obesity	8/57 (12)	3/24 (11)	>0.999
Bone metabolism disorder	19/46 (29)	2/25 (7)	0.028

**Table 2 jpm-13-00350-t002:** Overview of patient and injury-related factors of the retrospective cohort analysis.

	Group 1	Group 2	Group 3	*p*-Value ^#^
total number	35	92	3	
gender (male/female)	24 m, 11 f	45 m, 47 f	2 m, 1 f	*p* = 0.0469
age (years), mean SD	47.17 ± 16.51	55.57 ± 21.32	26.00 ± 6.56	*p* = 0.0438 *
follow up (months)				
multiple injured				
monotrauma	6	35	0	*p* < 0.001
at least one other fracture	16	19	0	
polytrauma (ISS > 16)	13	38	3	
fracture classification				
AO 33 A1	9	8	1	*p* < 0.0001
AO 33 A2		6		
AO 33 A3		12		
AO 33 B1	4	3	1	
AO 33 B2	9	2		
AO 33 B3	1			
AO 33 C1	1	4		
AO 33 C2		23	1	
AO 33 C3	6	26		
one-step surgery two-step surgery	22 13	37 55	0 3	*p* = 0.022
postoperative complications				
early (wound infection, postoperative hematoma, thromboembolism)	4	25	0	
late (secondary fracture dislocation, implant malalignment, delayed union)	4	42	0	*p* = 0.002
Kellgren–Lawrence increase 6 months post- vs. pre-surgery	0.1739 ± 0.3875	0.2553 ± 0.4207		0.584
Kellgren–Lawrence increase >1 year post- vs. pre-surgery	0.7500 ± 1.0733	0.7903 ± 0.7495		0.238
implant removal (after months (mean SD))	15 (19.06 ± 18.33)	26 (19.47 ± 7.03)	1 (23)	<0.999

^#^ For continuous values, Kruskal–Wallis ANOVA was used. For ordinal data, the chi-square test was used. * Significance between groups 2 and 3.

**Table 3 jpm-13-00350-t003:** Sub-analysis of correlation between fracture entity and postoperative complications.

AO-ASIF Classification	33A1-3 *n* = 26	33B1-3 *n* = 13	33C1-3 *n* = 53		*p*-Value Chi-Square
Wound healing disorder <14 d post-surgery	1/25	3/10	10/43		0.151
			C2 *n* = 22	C3 *n* = 25	
			1/21	7/18	0.032
Infections after >6 weeks <1 year post-surgery	1/12	2/5	8/32		0.130
			C2	C3	
			3/15	5/13	0.528
Delayed fracture healing: 3–9 months post-surgery	5/11	4/4	20/22		0.316
			C2	C3	
			8/10	12/7	0.462
Fracture non-union >9 months	3/10	2/6	20/21		0.049
			C2	C3	
			8/10	12/7	0.462

**Table 4 jpm-13-00350-t004:** Results of the measurement of the mechanical and anatomical axes and joint angles on full leg radiographs at latest follow up.

	Group 1	Group 2	Chi-Square
aMPFW (*n*)	10	25	
normal (80–89°)	9	13	*p* = 0.188
mild deformity (90–95° or 74–79°)	1	7	
pathologic deformity (>95°/<74°)	0	5	
mLPFW (*n*)	10	25	
normal (85–95°)	9	14	*p* = 0.254
mild deformity (79–84° or 96–101°)	1	6	
pathologic deformity (>101°/<79°)		5	
aLDFW (*n*)	10	25	
normal (79–83°)	9	10	*p* > 0.999
mild deformity (84–89° or 73–78°)	1	15	
pathologic deformity (>89°/<73°)	0	0	
mLDFW (*n*)	10	25	
normal (84–90°)	10	18	*p* > 0.999
mild deformity (91–96° or 78–83°)	0	7	
pathologic deformity (>96°/<78°)	0	0	
Leg length discrepancy (>1.5 cm, *n*)	0	4	n.s.

## Data Availability

Data are contained within the article.

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
