# Peer review of "Outcome Evaluation of Distal Femoral Fractures Following Surgical Management: A Retrospective Cohort Study"

_jpm, 2023, doi:10.3390/jpm13020350_

Round 1

Reviewer 1 Report

In the present study, different osteosynthesis procedures for the treatment of distal femoral fractures were compared. A differentiation was made between screw osteosynthesis, plate osteosynthesis and nailless osteosynthesis.  The endpoint of the retrospective study was the operative outcome as measured by wound healing, bone healing and secondary osteoarthritis.

The underlying idea of the study is very interesting and of clinical and scientific interest.

However, at the current time, the present methodology does not allow any conclusions to be drawn for clinical practice.

The comparison of the outcome is significantly influenced by the type of fracture and the clinical condition of the patient including comorbidities, which is not considered in this study. Even if the fracture type is listed, no subgroup analysis was performed, although this is obligatory for a final statement.

Even though I recommend to reject the present study due to the above mentioned reasons, I am sure that after a fundamental revision a study worth reading will result.

Author Response

Response to Reviewer 1 Comments

Dear reviewer 1,

Thank you for your time reviewing our manuscript. We appreciate your valuable comments and think they helped us to improve our manuscript.

Point 1: However, at the current time, the present methodology does not allow any conclusions to be drawn for clinical practice.

Response 1: We have revised the methodology and added further subgroup analysis to allow conclusions for the clinical practice.

Point 2: The comparison of the outcome is significantly influenced by the type of fracture and the clinical condition of the patient including comorbidities, which is not considered in this study. Even if the fracture type is listed, no subgroup analysis was performed, although this is obligatory for a final statement.

Response 2: We have made further analysis and did include relevant comorbidities of the included patients to allow potential influence of risk factors on postoperative complications and osseous union. Additionally we have reviewed the single fracture types, if there was a relevance of the sustained fracture entity on the postoperative course. The results of both analysis are now included in the manuscript.

Reviewer 2 Report

Fractures of the distal femur (the far end of the thigh bone just above the knee) are a considerable cause of morbidity. Various different surgical and non-surgical treatments have been used in the management of these injuries but the best treatment remains unknown.The group number 3 is with 3 patients very small group and not significant the results.

 An approach of primary definitive fixation with condylar locking plate and antibiotic impregnated collagen sheet secondary to early aggressive debridement in open distal femur fractures shows significant results in terms of functional and radiological outcomes with minimal complications.

The references are incomplete missing age 2020.2021 and 2022.

  They should reorganise the study group because the number 3 group hase onl;y 3 patient.They can be replace with mega knee ptosthesis,mono-or poliaxyal plate.

  So they need revision in papaer.

Author Response

Response to Reviewer 2 Comments

Dear reviewer 2

Thank you for your time reviewing our manuscript. We appreciate your valuable comments and think they helped us to improve our manuscript.

Point 1: Fractures of the distal femur (the far end of the thigh bone just above the knee) are a considerable cause of morbidity. Various different surgical and non-surgical treatments have been used in the management of these injuries but the best treatment remains unknown.The group number 3 is with 3 patients very small group and not significant the results.

Response 1: We are aware that the number of included patients to nb group 3 is far to low to allow statistical analysis. We have mentioned that in the paper (methods and limitations section). But we did want to mention this group as intramedullary nailing systems are a relevant treatment option in the management of distal femur fractures.

Point 2: An approach of primary definitive fixation with condylar locking plate and antibiotic impregnated collagen sheet secondary to early aggressive debridement in open distal femur fractures shows significant results in terms of functional and radiological outcomes with minimal complications.

Response 2: Thank you for this input. We did review the latest literature and have now discussed these management options in the paper.

Point 3: The references are incomplete missing age 2020.2021 and 2022.

Response 3: We have added references from the years 2020 – 2022.

Point 4: They should reorganise the study group because the number 3 group hase onl;y 3 patient.They can be replace with mega knee ptosthesis,mono-or poliaxyal plate.

Response 4: As you mentioned correctly group 3 is too small for statistical analysis. With respect to the given time of 8 days for this revision we were not able to replace this group by another cohort. Moreover megaprothesis are not performed at our institution in trauma cases but are a reserved method in the management of cancer patient. But we have discussed the treatment option of megaprothesis in the paper. Please accept our response and explanation.

Round 2

Reviewer 1 Report

A subgroup analysis depending on the AO classification was added to the current version. The conclusions drawn from this are adequately presented in the discussion and conclusion. For the future, a more in-depth analysis would be desirable, comparing the outcome of the same fracture types depending on the osteosynthesis procedure.